# Anti-SARS-CoV-2 S-RBD IgG Antibody Responses after COVID-19 mRNA Vaccine in the Chronic Disorder of Consciousness: A Pilot Study

**DOI:** 10.3390/jcm10245830

**Published:** 2021-12-13

**Authors:** Maria Elena Pugliese, Riccardo Battaglia, Antonio Cerasa, Maria Girolama Raso, Francesco Coschignano, Angela Pagliuso, Roberta Bruschetta, Giovanni Pugliese, Paolo Scola, Paolo Tonin

**Affiliations:** 1Intensive Rehabilitation Unit, S’Anna Institute, 88900 Crotone, Italy; r.battaglia@isakr.it (R.B.); antonio.cerasa@irib.cnr.it (A.C.); m.raso@istitutosantanna.it (M.G.R.); f.coschignano@isakr.it (F.C.); a.pagliuso@isakr.it (A.P.); g.pugliese@isakr.it (G.P.); p.scola@isakr.it (P.S.); patonin18@gmail.com (P.T.); 2Institute for Biomedical Research and Innovation (IRIB), National Research Council of Italy, 98164 Messina, Italy; roberta.bruschetta@irib.cnr.it; 3Pharmacotechnology Documentation and Transfer Unit, Preclinical and Translational Pharmacology, Department of Pharmacy, Health Science and Nutrition, University of Calabria, 87036 Rende, Italy

**Keywords:** COVID-19, disorder of consciousness, vaccination, antibody responses

## Abstract

Objective: In the last year, a large amount of research has investigated the anti-spike protein receptor-binding domain (S-RBD) antibody responses in patients at high risk of developing severe acute respiratory syndrome because of COVID-19 infection. However, no data are available on the chronic disorder of consciousness (DOC). Methods: Here, we evaluated anti-S-RBD IgG levels after vaccination in chronic DOC patients compared with demographically matched healthy controls (HC) by indirect chemiluminescence immunoassay. All individuals completed a two-dose-cycle vaccination with Pfizer mRNA vaccine (BNT162b2), and antibody responses were evaluated at 30 and 180 days after the administration of the second dose of vaccination. Results: We compared 32 DOC patients with 34 demographically matched healthy controls. Both DOC and HC groups showed a similar antibody response at 30 days, whereas at follow-up (180 days) DOC patients were characterized by lower S-RBD IgG levels with respect to controls. Additional multiple regression analyses including demographical and clinical comorbidities as predictors revealed that age was the only factor associated with the decrease in S-RBD IgG levels at follow-up (180 days). Elderly individuals of both groups were characterized by a reduction in the antibody responses with respect to younger individuals. Conclusions: Our results show an efficacy seroconversion in DOC patients in the first period after vaccination, which significantly declines over time with respect to healthy controls.

## 1. Introduction

The eruption of the COVID-19 pandemic, caused by the newly discovered SARS-CoV-2 virus, has had a profound impact on human life on a global scale [1,2]. COVID-19 has affected and still affects millions of people worldwide, resulting in high mortality and morbidity rates as well as high healthcare costs and difficulties in treatment [3,4].

Since January 2020, when the first sequencing of SARS-CoV-2 became public, the scientific community has worked toward the rapid development of mRNA, protein, viral vector, and other types of COVID-19 vaccines. Currently, vaccines authorized in the European Union (EU) to prevent COVID-19 are distinct in genetic (Pfizer–Biontech and Moderna) [5,6] or viral vector (Janssen and Vaxzevria). The first vaccine given such authorization was an mRNA in lipid nanoparticles (LNPs), from Pfizer–BioNTech. Findings from studies conducted in the United States and Germany among healthy men and women showed that two 30 μg doses of Pfizer vaccine (BNT162b2) elicited high SARS-CoV-2 antibody titers and robust antigen-specific T-cell responses (CD8+ and Th1-type CD4+). BNT162b2 was 95% effective in preventing COVID-19 at 7 days after the second dose [6]. Individuals with underlying autoimmune disorders and those on immune-modulatory therapies were not included in these early trials, nor were severely impaired neurological patients.

To address this gap, a few recent studies investigated the response to the COVID-19 vaccine in immunocompromised patients, such as solid-organ transplant recipients, patients with hematological malignancies, cancer, end-stage renal disease and rheumatologic patients on immunotherapy, by measuring SARS-CoV-2 IgG production after they had been vaccinated. The conclusion was a suboptimal response with a high percentage of non-responder patients that raised the concern of ongoing risk of COVID-19 despite vaccination in immunocompromised hosts and the need of a third dose of vaccination [7,8,9,10,11,12].

Chronic disorders of consciousness (DOC) are neurological conditions characterized by severe alterations in the level of consciousness including vegetative state (VS) and the minimally conscious state (MCS). To date, the long-term effects of brain injury on the immune system are unknown and very few data are available on DOC immunocompetence. A laboratory study conducted on mice suggests a chronic and persistent innate and adaptive immune dysfunction after traumatic brain injury, with changes in multiple leukocyte subsets and characterized by acute immune suppression and chronic immune impairment [13]. In the clinical setting, Munno et al. [14] showed a profound impairment of phagocytosis and killing of monocytes and low serum levels of IFNy in vegetative state patients. In another study, Satzbon et al., [15] investigated 11 post-traumatic persistent vegetative-state patients who had moderate to severe pyogenic infections. They found an alteration of the humoral immunity, with a consequently defective opsonization and a neutrophil dysfunction in 27% of patients [15]. However, they were unable to determine if these impairments were the result of the primary insult or a complication of prolonged unconsciousness. To resolve this question, they designed a second study [16] to investigate the immunological changes that occur shortly after severe brain injury. Significant deficiencies of the immune system, particularly the cellular arm, were precipitated by severe brain injury within 72 h of the event (43% of patients had humoral defects and 79% had cellular defects).

In the era of COVID-19, it is mandatory to identify vulnerable people at high risk of developing chronic immune impairments and complications that could lead to mortality. This preliminary evidence would suggest that DOC patients could be a candidate phenotype since data on immune responses to vaccines in this patient cohort are particularly scarce. For this reason, as already performed for other categories of immunocompromised patients [7,8,9,10,11,12], we evaluated, for the first time, the amount of SARS-CoV-2 S-RBD IgG titer after a complete two-dose-cycle COVID-19 vaccination in chronic DOC patients with respect to a demographically matched healthy control group.

## 2. Materials and Methods

### 2.1. Enrollment

Among 42 consecutive patients admitted to the long-term care facility of the S. Anna Institute (Crotone, Italy), between January 2021 and August 2021, the patients fitting with the following inclusion criteria were enrolled: (a) age > 18 years; (b) clinical diagnosis at the admission of vegetative state (VS) or minimally conscious state (MCS), according to standard diagnostic criteria [17]; (c) patients affected by DOC for at least 12 months; (d) complete two-dose-cycle vaccination with BNT162b2. Exclusion criteria were: (a) medically confirmed COVID-19 before and during protocol; (b) administration of only one dose of vaccination because of adverse reaction to the vaccine, or discharge/death; (c) deferring the second dose of vaccination due to clinical instability (i.e., severe infection). All patients were monitored with a monthly antigenic nasal swab for screening purposes.

Thirty-four healthy volunteers with no previous history of neurological, psychiatric or immunological diseases and without COVID-19 infection were matched for age, gender and education with enrolled DOC patients. The demographic and clinical characteristics of all participants are summarized in Table 1.

### 2.2. Ethics

The study was approved by the Ethical Committee of the Regione Calabria (N protocol n. 166, 15 July 2021), according to the Helsinki Declaration. Written informed consent was obtained from the legal representative of each patient.

### 2.3. Procedure

All subjects (DOC and controls) received two doses of Pfizer–BioNTech vaccine (Pfizer Inc., New York, NY; BioNTech SE, Mainz, Germany) 21 days apart. In all vaccinated subjects, anti-S-RBD IgG levels were exactly measured at the same time: 30 days and after six months (follow-up), after completing the vaccination cycle with BNT162b2 vaccine.

### 2.4. Biochemical Analysis

Four weeks after the second vaccine dose, we performed a panel of standard blood analysis, including complete blood count, reactive protein C, urea, creatinine and electrolytes and, at the same time, we dosed the antibody IgG anti-SarsCov19 titer (S1-RBD) in both groups. After 6 months, we dosed again the antibody titers to evaluate their variation over time and the difference between the two groups. In this study, a commercially available immunoassay was used, the anti-SARS-CoV-2 S-RBD IgG (Snibe Diagnostics, New Industries Biomedical Engineering Co., Shenzhen, China). SARS-CoV-2 S-RBD IgG is a chemiluminescent immunoassay (CLIA) that determines IgG Ab against the RBD of the Spike (S) protein of the virus, in human serum or plasma. All analyses were performed on Maglumi 800 (Snibe Diagnostics, Shenzhen, China), with results expressed in arbitrary units. The upper limit of the method without sample dilution is 100 AU/L and the cut-off for reactivity is 1 AU/L. Correction factor versus the international standard WHO 20/136 was fixed (4.33 for Maglumi). After correction, the linear range of this test is from 1 to 433 BAU/mL. Negative results are thus depicted as 0.99 BAU/mL, and highly reactive samples are capped with a value of 433 BAU/mL in this study.

### 2.5. Statistical Analysis

Statistical analyses were performed by SPSS statistical software v.17.0 (SPSS Inc., Chicago, IL, USA) and R Language v.4.0.3 (R Foundation for Statistical Computing, Vienna, Austria). Normality distribution was assessed preliminarily by q-q plot, and Kolmogorov–Smirnov tests. Quantitative variables were expressed by the median values, while categorical variables by relative frequency. Differences between groups for continuous and categorical variables were estimated, respectively, by non-parametric Kruskal–Wallis test (if >2 groups) and Mann–Whitney U-test (with Bonferroni’s correction when needed).

To identify factors associated with antibody responses, univariate and multivariate linear regression analyses were performed. In the univariate analysis we used Mann–Whitney U-test including all categorical variables. In the multivariate regression analysis we included demographical data (age and sex) and clinical comorbidities (mainly impacting clinical course and immunocompetence of DOC patients: heart disease, diabetes mellitus, heart disease, tumors, chronic obstructive pulmonary disease) as independent variables and antibody response as dependent variables. In the regression model, age was considered as a continuous variable, whereas all the rest of the variables were categorical. Patients and controls who did not have any response at 4 weeks were not included in the model. The R-squared and the ANOVA tests were used to assess the adequacy of the models. All statistical analyses had α levels of < 0.05 for defining significance

## 3. Results

Thirty-two DOC patients (62.5% in MCS, 37.5% in VS) were enrolled in this protocol. The enrolled patients were characterized by vascular (37.5%), traumatic (31.25%), anoxic events (18.75%) and other etiologies (12.5%). Patients were demographically matched with a healthy control group (Table 1).

After the first vaccine dose, none of the DOC and HC individuals developed severe adverse effects (Table 2). Reported side effects were mostly mild to moderate and short-lasting. They included: fever, fatigue, headache, muscle pain, chills, diarrhea, and pain at the injection site. A high percentage of healthy subjects reported local pain (73.5%), some developed fever (11.7%) and about half of the control group reported diffuse muscle pain, headache or fatigue (47%). DOC patients showed very few side effects, with only one patient with high fever and a case of enlarged lymph nodes. After the second dose local symptoms were less prevalent (64.7%) while mild systemic adverse reactions were more prevalent (HC fever 14.7%, muscle pain and fatigue 55.8%; DOC fever 6.25%). We had no cases of respiratory distress, seizures or death.

Anti-RBD IgG response was evaluated in both groups 30 and 180 days from the second vaccine dose. After the first measurement we did not detect any significant difference in the anti-S-RBD IgG levels between groups (DOC: 433 (99–433) BAU/mL; HC: 433 (156.5–433) BAU/mL) (K-W = 1.52; *p*-level = 0.217). At follow-up (180 days), DOC patients were characterized by lower levels of antibodies compared to controls (DOC: 16.8 (9.2–57.4) BAU/mL; HC: 39 (15.6–246.6) BAU/mL) (K-W = 11.7; *p*-level < 0.001). Figure 1 shows the distribution of antibody responses over timepoints in the two groups.

To assess which variables are associated with the detected decline in antibody responses, we performed two additional univariate and multivariate analyses (Table 3). Univariate analysis revealed that the variable “group” is the only one significantly associated with the different levels of antibody responses detected at 6 months. Multiple regression analysis revealed an additional effect of the “age” variable on the decrease in antibody responses. Indeed, pooling all subjects together, we found that the highest age corresponded to the lowest antibody responses (R = 0.51; R^2^ = 0.25; F-value = 2.83; *p*-value = 0.01) (Table 3) (Figure 2A). We also repeated this analysis within every single group, but no significant results were found. In other words, age impacts antibody response similarly in the controls as well as in DOC patients (Figure 2B).

## 4. Discussion

The evaluation of SARS-CoV-2 S-RBD IgG titers represents a useful tool to estimate the individual protection against SARS-CoV-2 infection. It is noteworthy that the evaluation of antibodies against S-RBD IgG is the most important to assess the protection against SARS-CoV-2 infection due to their seroconversion activity. However, since SARS-CoV-2 is a newly emerging virus, the antibody responses in COVID-19 patients and, especially, in vaccinated subjects remain largely unknown. In our study, after a complete COVID-19 vaccination, DOC patients showed lower levels of antibodies with respect to HC more evident after 6 months. An additional aging linear effect on antibody responses was also revealed in both groups, since antibody levels were lower in older than younger individuals. Our findings are partially in accordance with Muller et al., [18] who showed that the elderly had significantly lower levels of antibodies than young subjects. In a large observational study, Lo Sasso et al. observed an efficacy antibody response after vaccination with age- and time-related differences [15]. Instead, we did not detect differences between males and females, although in other studies higher antibody titers in female subjects were reported [19,20].

Although it is known that an acute and severe brain injury leads to an altered immune system response, what happens to chronic DOC patients’ immune system functioning is largely unknown. Almost everything we know is derived from studies on traumatic brain injury (TBI) patients in the acute phase. In addition to the brain injury itself, it is increasingly highlighted that a TBI may alter the systemic immune response in a way that makes TBI patients more vulnerable to infections in the acute post-injury period [21,22,23,24,25,26,27,28,29,30]. In the clinical setting, few and contradictory data are available on very small cohorts of vegetative-state patients [13,14,15,16,30]. To our knowledge, no data are available on specific or acquired immunity in chronic DOC patients.

For this reason, we decide to evaluate the post-vaccination IgG titer as a tool to indirectly investigate immune activity and reactivity in this group of patients. Thus, testing the efficacy of COVID-19 vaccination on DOC patients has two main merits: (a) demonstrate the presence of adaptive responses in a group of patients traditionally considered as immunocompromised, because of the onset clinical severity, the high frequency of complications due to neurosurgery, infections and being long-term bedridden; and (b) stimulate further studies with larger samples to investigate if the detected pattern of anti-S-RBD IgG levels is similar in stroke, traumatic or anoxic patients.

### Limitations

The main limitation of this study is the sample size. Despite a perfect matching in demographics between groups at baseline, we recognized that an evaluation on larger samples should be performed in order to confirm our preliminary findings. However, we should bear in mind that, generally, studies on long-term chronic DOC patients are traditionally carried out on very few patients, often only a few units. In this view, we believe that the observations that emerged from this study are notable for further studies.

Another important limitation is the antibody responses of patients that could exceed the linear range of the Snibe Diagnostics Anti-SARS-CoV-2 S test, which is capped with a value of 433 BAU/mL (ceiling effect). This is a commonly reported issue in COVID-19-related studies [31]. We suggest for future studies to determine exact values of BAU/mL if these exceed the linear range of the antibodies test using a 1:10 dilution of the sample that can be measured subsequently.

## 5. Conclusions

Our pilot study shows an efficient antibody response after vaccination in DOC patients, although a significantly greater decrease was observed with respect to controls at 6 months. This observation could be useful in assisting decision making when choosing to administer additional vaccine doses for this at-risk category.

## Figures and Tables

**Figure 1 jcm-10-05830-f001:**
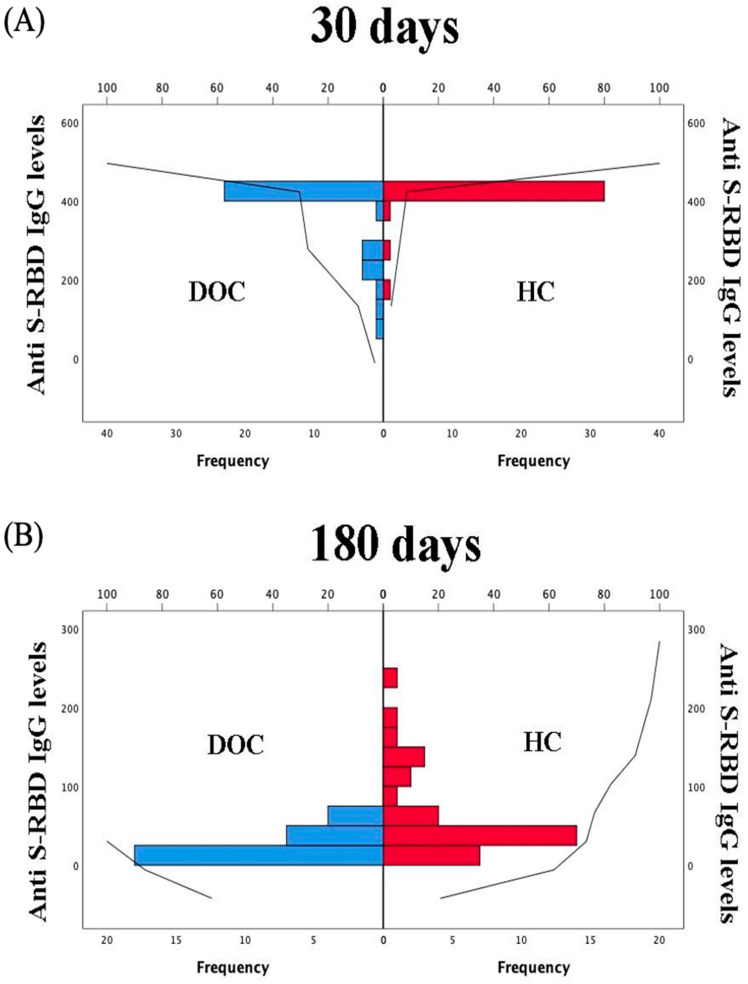
Frequency distribution of anti-S-RBD IgG levels in the two groups after complete vaccination over time at 30 days (**A**) and 180 days (**B**).

**Figure 2 jcm-10-05830-f002:**
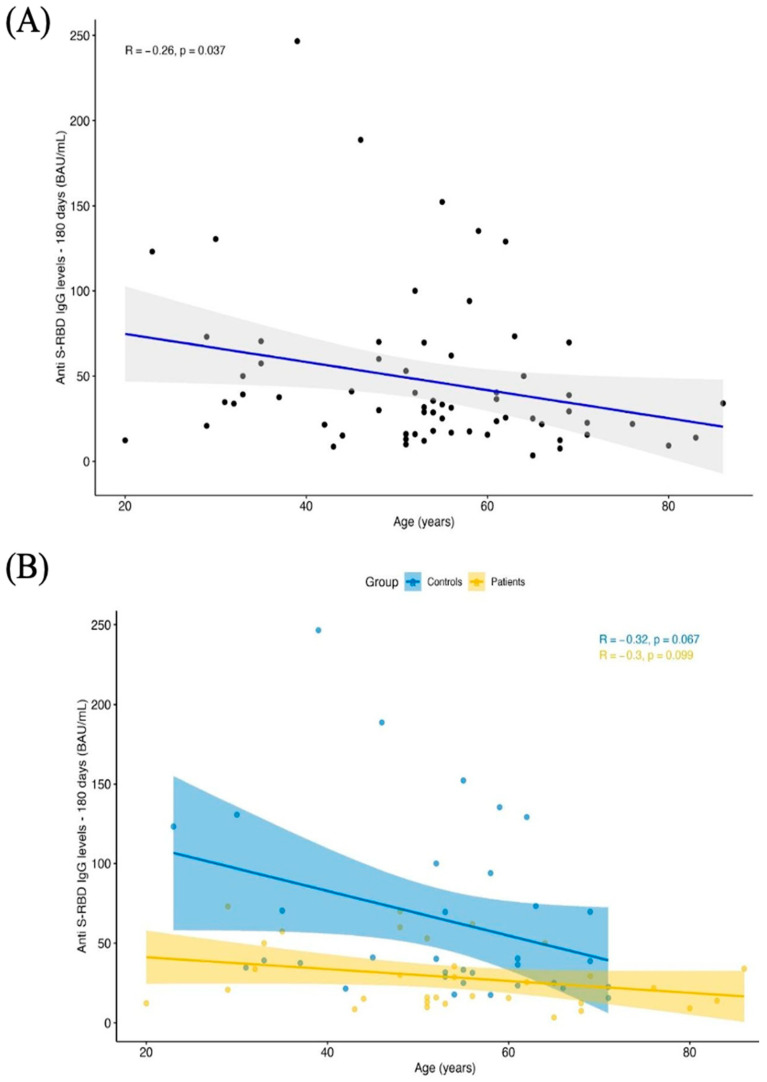
Regression analysis showing the relationship between age factor and antibody responses in the whole group of enrolled subjects (**A**) and within every single group (**B**).

**Table 1 jcm-10-05830-t001:** Demographic and clinical characteristics of enrolled DOC patients and controls.

	DOC (n = 32)	HC (n = 34)	p-Level
Age	53.5 ± 15.8	52.9 ± 12.5	0.819 ^§^
Sex, (%) male	53%	53%	1.00 *
Hypertension (Yes; %)	43.7%	14.7%	0.02 *
Diabetes mellitus (Yes; %)	0%	2.9%	0.97 *
Heart disease (Yes; %)	18.7%	8.9%	0.41 *
Renal insufficiency (Yes; %)	0%	2.9%	0.97 *
Obstructive pulmonary disease (Yes; %)	9.3%	5.8%	0.94 *
Liver disease (Yes; %)	6.2%	0%	0.47 *
Endocrinopathies (Yes; %)	9.3%	8.8%	0.72 *
Tumor (Yes; %)	3.1%	0%	0.97 *
CRS-r at enrolment	8.3 ± 3.6		
Time from injury (years)	4.1 ± 3.6		
Etiology n (%) Vascular			
	12 (37.5%)		
Traumatic	10 (31.25%)		
Anoxic	6 (18.7%)		
Others (dementia, infections/post-surgery)	4 (12.5%)		
Diagnosis n (%) VS MCS			
	12 (37.5%)		
	20 (62.5%)		

VS: vegetative state; MCS: minimally conscious state. CRS: Coma Recovery Scale—Revised. ^§^ Kruskal–Wallis test; * Chi2 test.

**Table 2 jcm-10-05830-t002:** Adverse events after the first or second dose of vaccine.

	DOC Patients		Healthy Controls	
	After First Dose	After Second Dose	After First Dose	After Second Dose
Local pain	0%	0%	73.5%	64.7%
Fever	3.1%	6.25%	11.7%	14.7%
Diffuse muscle pain, headache, fatigue	0%	0%	47%	55.8%
Severe reactions	0%	0%	0%	0%

**Table 3 jcm-10-05830-t003:** Univariate and multivariate analyses to evaluate the factors influencing the decline in antibody responses at 6 months from vaccination.

Univariate Analysis (Mann–Whitney U)	Statistic	*p*-Level
Group (DOC vs. HC)	263	< 0.001
Sex	510	0.699
Hypertension	333	0.110
Diabetes mellitus	27.0	0.793
Heart disease	222	0.519
Renal insufficiency	22.0	0.600
Obstructive pulmonary disease	130	0.586
Liver disease	53.0	0.694
Endocrinopathies	162	0.696
Tumor	6.0	0.172
**Multivariate Analysis (Omnibus ANOVA Test)**	**Statistic**	***p*-Level**
Age	6.6660	0.012
Group (DOC vs. HC)	11.1537	0.001
Sex	0.9258	0.34
Diabetes mellitus	0.1570	0.693
Heart disease	0.9867	0.325
Tumors	1.1026	0.298
Chronic obstructive pulmonary disease	0.0940	0.768

## Data Availability

The data presented in this study are available on request from the corresponding author.

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
