# Peer review of "Anti-SARS-CoV-2 S-RBD IgG Antibody Responses after COVID-19 mRNA Vaccine in the Chronic Disorder of Consciousness: A Pilot Study"

_jcm, 2021, doi:10.3390/jcm10245830_

Round 1

Reviewer 1 Report

THe authors carried out a study comparing  the amount of IgG neutralizing 63 antibody response after a complete two-dose cycle COVID-19 vaccination (BNT162b2, Pfizer bioNtech) between  42 consecutive DOC patients admitted for to the long-term care facility of the S. Anna 68 Institute (Crotone, Italy),from January 2021 to August 2021,  with 34 well-match healthy volunteers . *The ethics  procedure are  correct.

Methods  results  are clearly exposed , concluding to an absence of difference  between the both groups regardind the primary endpoint ( 1-month amount of neutralizing antibody IgG anti-SarsCov19 titer (S1- 96 RBD). and but  a lower one at 6-month  explaining  by a difference of age between groups.

Then manuscript  is well done however some comments are  needed:

MEthods :   

  • the choice of neutralizing antibody IgG anti-SarsCov19 titer (S1- 96 RBD) seems right  to control the immunologival repsonse to the Vaccine 
  • we do not know  whether Healthy control have been vaccinated  at the same period of time.
  •  

Results 

We  do not find any comparison of both populations  at baseline ( comorbidities, Age, sex etc...) It lacks in a such  pilot study

They  did not define what is young  and old when comparing 2 new groups  that have not been stated previously in method  part.

LImitations :

  • The  fact that authors stated they matched population and find a  difference explaning by age is  a problem.
  • the statistical  analysis  to explain difference at month 6  is not described ;  a comparison of all  variables collected explaining potentially the difference have to be shown ; 
  • they should realize a univariate and  multivariate analysis to explore all ( at least with all variable associated with lower vaccine response)  and not only age.
  • the background of the study is not very well supported by scientfic literature:  we do not  know why  Patient may have lower vaccine response. there is no  real studies on other vaccines  but  vaccines are done without  any questions.
  • the number od patients is limited but the DOC patients are rare and represent a special population

to conclude  the study is  well done (except for charaterisitic of both populations)  HOwever the main limitation is the scientific background of the study   that i receive not very well. To improve that point authors should give other data regarding other  vaccines  to convice us of the need of the study.

Author Response

THe authors carried out a study comparing  the amount of IgG neutralizing 63 antibody response after a complete two-dose cycle COVID-19 vaccination (BNT162b2, Pfizer bioNtech) between  42 consecutive DOC patients admitted for to the long-term care facility of the S. Anna 68 Institute (Crotone, Italy),from January 2021 to August 2021,  with 34 well-match healthy volunteers . *The ethics  procedure are  correct.

Methods  results  are clearly exposed , concluding to an absence of difference  between the both groups regardind the primary endpoint ( 1-month amount of neutralizing antibody IgG anti-SarsCov19 titer (S1- 96 RBD). and but  a lower one at 6-month  explaining  by a difference of age between groups.

Then manuscript  is well done however some comments are  needed:

Methods :   

  1. the choice of neutralizing antibody IgG anti-SarsCov19 titer (S1- 96 RBD) seems right  to control the immunologival repsonse to the Vaccine we do not know  whether Healthy control have been vaccinated  at the same period of time.

REPLY: Vaccination was carried out on the same day for patients and the healthy control group, both for the first dose and second dose. In the same way, antibody titer evaluation was carried out on the same day for both groups, 30 and 180 days after complete vaccination. This additional information has been included in the main text

Results 

  • We  do not find any comparison of both populations  at baseline ( comorbidities, Age, sex etc...) It lacks in a such  pilot study

REPLY: Table 1 has been changed accordingly. Moreover, please consider that we now added additional information about clinical comorbidities in both groups.

  • They  did not define what is young  and old when comparing 2 new groups  that have not been stated previously in method  part.

REPLY: In the statistical section, we better describe that age was considered as a continuous variable.

LImitations :

  • The  fact that authors stated they matched population and find a  difference explaning by age is  a problem.

REPLY: As said in the next points, moving from univariate to multivariate analysis, with the inclusion of multiple regression model we never detected a different effect of age as a function of the group, but a similar age-dependent effect on antibodies response was found in both groups.

  • the statistical  analysis  to explain difference at month 6  is not described ;  a comparison of all  variables collected explaining potentially the difference have to be shown ; 

REPLY: We used the parametric Kruskal-Wallis test to evaluate the presence of statistical differences in antibody responses at follow-up between the two groups. Overall antibodies response distribution for the two groups has been shown in Figure 1, moreover, descriptive data about antibodies response have been reported in the main text. We believe that all relevant information has been provided.

  • They should realize a univariate and  multivariate analysis to explore all ( at least with all variable associated with lower vaccine response)  and not only age.

REPLY: We would like to thank this reviewer for this important suggestion. We now performed multiple regression analysis considering all demographical and clinical data that could affect the detected decreasing of antibodies response. As you can see (Table 3) the effect of age is significantly relevant only when patients and controls are pooled together. We did not reveal a different age effect as a function of the group. For this reason, we modified the main text accordingly to this new finding and we would like to thank again this reviewer for helping us to improve our study. 

  • the background of the study is not very well supported by scientfic literature:  we do not  know why  Patient may have lower vaccine response. there is no  real studies on other vaccines  but  vaccines are done without  any questions. The number od patients is limited but the DOC patients are rare and represent a special populationto to conclude  the study is  well done (except for charaterisitic of both populations)  HOwever the main limitation is the scientific background of the study   that i receive not very well. To improve that point authors should give other data regarding other  vaccines  to convice us of the need of the study.

REPLY: The Introduction has been completely re-formulated following the reviewer’s suggestion. Please consider that due to the nature of this neurological disorder, obviously, data on immune responses to vaccines in this patient’s cohort are particularly scarce. The idea underlying this study is that DOC represents another category of immunocompromised patients, such as solid organ transplant recipients, patients with hematological malignancies, cancer, end-stage renal disease, rheumatologic patients on immunotherapy, which are at high risk to develop chronic immune impairments and complications that may lead to lethality if they have been in contact with COVID-19 virus. For this reason, since in all previous clinical disorders the amount of SARS-CoV-2 IgG response after a complete two-dose cycle COVID-19 vaccination has been investigated, we tried to replicate this investigation also in this vulnerable chronic long-term debilitating disease.

Please also consider that there are data regarding other vaccines including HBV and Influenza Virus post-vaccine seroconversion. Here, antibody levels are checked to evaluate vaccine response and long-term immunity:

  • (Farooq PD et al. Hepatitis B Vaccination and Waning Hepatitis B Immunity in Persons Living with HIV. Curr HIV/AIDS Rep. 2019 Oct;16(5):395-403. doi: 10.1007/s11904-019-00461-6. PMID: 31468298;
  • Steketee RW e al. Seroresponse to hepatitis B vaccine in patients and staff of renal dialysis centers, Wisconsin. Am J Epidemiol. 1988 Apr;127(4):772-82. doi: 10.1093/oxfordjournals.aje.a114858. PMID: 2965510;
  • Beyer et al., 1989. Antibody induction by influenza vaccines in the elderly: a review of the literature. Vaccine 7, 385–394;
  • Beyer, et al 2002. Cold-adapted live influenza vaccine versus inactivated vaccine: systemic vaccine reactions, local and systemic antibody response, and vaccine efficacy. A meta-analysis. Vaccine 20, 1340–1353
  • Palache et al., influenza vaccines - the effect of vaccine dose on antibody-response in primed populations during the ongoing interpandemic period - a review of the literature, Vaccine, 11(9), 1993, pp. 892-908).

Regarding SARS-Cov2 Vaccine Seroconversion, the clinical implications of the serology test and the presence of antibodies and their levels remain to be fully clarified. Nevertheless, there are several reports regarding the correlation of antibodies to SARS-CoV-2 (Lumley SF et al. Oxford University Hospitals Staff Testing Group: Antibody status and incidence of SARS-CoV-2 infection in health care workers. N Engl J Med 384: 533–540, 2021; Earle KA, Ambrosino DM, Fiore-Gartland A, Goldblatt D, Gilbert PB, Siber GR, Dull P, Plotkin SA: Evidence for antibody as a protective correlate for COVID-19 vaccines. Vaccine, 2021).

Reviewer 2 Report

The study by Pugliese and her colleagues investigated the anti-RBD antibodies (Ab) response elicited by the BNT162b2 mRNA vaccine in chronic disorder of consciousness (DOC) patients. They matched 32 DOC patients with 34 healthy controls (HC). They showed a similar Ab response in DOC patients and HC 30 days after the second vaccine injection but a decline of anti-RBD Ab levels in DOC patients in comparison with HC at 180 days, especially in elderly patients.

Despite an interesting immunogenicity study among DOC patients, for which there is a lake of data, I have some concerns about this work.

Major points:

- In the manuscript, the authors should remove all the “neutralizing” mention. Indeed, in their study, they used a serological assay (SARS-CoV-2 S-RBD IgG, Snibe Diagnostics) to quantify anti-RBD IgG and not a functional quantification of neutralizing Ab (i.e virus neutralization test, pseudo-neutralization assay or ELISA RBD-ACE2 blocking assay). So, it appears uncertain to discuss about Neutralizing Ab in this study.

- The upper limit of detection of the serological test is 433 BAU/ml. This limit appears quite low because it has been demonstrated that the antibody response could be much higher. With this upper limit detection, the authors lose information about the magnitude of Ab response which could modify their conclusions. Indeed, what if the median Ab level of DOC patients is 500 BAU/ml and 2000 BAU for HC ? It seems interesting to perform dilutions of the participants sera to precisely determine the Ab levels. This point could explain why the authors have no difference between gender in Ab level responses than other published studies  (line 173-175).

Minor points:

- In the “Materials and Methods” section, the authors should directly express the unit of the serological assay in BAU/ml accordingly to the WHO international recommendations.

- Did the DOC patients have underlying medical conditions other than their consciousness impairment ?- In the table 2 and “Results” section, comparison of adverse events between DOC patients and HC should only rely on objective clinical signs than subjective signs (local pain, diffuse muscle pain, headache, fatigue) which seems difficult to be precisely determine in DOC patients. - In the “Discussion” section, line 163-164, the author assertion should be modified because anti-RBD IgG levels correlates with neutralizing Ab but not all anti-RBD IgG are necessarily neutralizing, and all neutralizing Ab are not necessarily anti-RBD IgG (i.e. IgA, anti-S2 IgG). It is important to clearly distinguish quantification of antibodies and assessment of their functionality (i.e. neutralization).

Author Response

The study by Pugliese and her colleagues investigated the anti-RBD antibodies (Ab) response elicited by the BNT162b2 mRNA vaccine in chronic disorder of consciousness (DOC) patients. They matched 32 DOC patients with 34 healthy controls (HC). They showed a similar Ab response in DOC patients and HC 30 days after the second vaccine injection but a decline of anti-RBD Ab levels in DOC patients in comparison with HC at 180 days, especially in elderly patients.

Despite an interesting immunogenicity study among DOC patients, for which there is a lake of data, I have some concerns about this work.

Major points:

- In the manuscript, the authors should remove all the “neutralizing” mention. Indeed, in their study, they used a serological assay (SARS-CoV-2 S-RBD IgG, Snibe Diagnostics) to quantify anti-RBD IgG and not a functional quantification of neutralizing Ab (i.e virus neutralization test, pseudo-neutralization assay or ELISA RBD-ACE2 blocking assay). So, it appears uncertain to discuss about Neutralizing Ab in this study.

REPLY: we would like to thank this reviewer for highlighting this point. We now generically speak about SARS-CoV-2 S-RBD IgG titer or seroconversion after SARS-Cov2 vaccination.

- The upper limit of detection of the serological test is 433 BAU/ml. This limit appears quite low because it has been demonstrated that the antibody response could be much higher. With this upper limit detection, the authors lose information about the magnitude of Ab response which could modify their conclusions. Indeed, what if the median Ab level of DOC patients is 500 BAU/ml and 2000 BAU for HC ? It seems interesting to perform dilutions of the participants sera to precisely determine the Ab levels. This point could explain why the authors have no difference between gender in Ab level responses than other published studies  (line 173-175).

  • REPLY: We completely agree with this reviewer. This is a commonly reported issue in COVID-19 related studies. We now recognized this limitation in a new section of the discussion. Overall, we cannot overcome this limitation, because of the unavailability of the samples stored in the molecular biology laboratories during the very early and critical phase of the Italian vaccination campaign.

As correctly affirmed by this reviewer, this is a limit for the first assay causing a “ceiling-effect”. Nevertheless, we were able to obtain very interesting clinical observations, such as the fact that all patients had antibody titers > 1 AU/l. Thus demonstrating that an immune response was developed, exceeding the reactivity cut-off to be declared ‘reactive’ for SARS-CoV-2 S-RBD IgG response.

As regard gender effect, we now included a new multivariate multiple regression analysis to better evaluate which variables might affect the detected decreasing of antibodies response. This additional analysis confirmed that gender is not associated with this effect. We believe that this lack of significance could be related to sample size.

Minor points:

- In the “Materials and Methods” section, the authors should directly express the unit of the serological assay in BAU/ml accordingly to the WHO international recommendations.

REPLY: Done

- Did the DOC patients have underlying medical conditions other than their consciousness impairment ?- In the table 2 and “Results” section, comparison of adverse events between DOC patients and HC should only rely on objective clinical signs than subjective signs (local pain, diffuse muscle pain, headache, fatigue) which seems difficult to be precisely determine in DOC patients.

  • REPLY: Following the reviewer’s suggestion we now included in Table 1 new information about underlying medical conditions in patients as well as in controls

- In the “Discussion” section, line 163-164, the author assertion should be modified because anti-RBD IgG levels correlates with neutralizing Ab but not all anti-RBD IgG are necessarily neutralizing, and all neutralizing Ab are not necessarily anti-RBD IgG (i.e. IgA, anti-S2 IgG). It is important to clearly distinguish quantification of antibodies and assessment of their functionality (i.e. neutralization).

REPLY: As said before we now generically speak about SARS-CoV-2 S-RBD IgG titer or seroconversion after SARS-Cov2 vaccination

Round 2

Reviewer 1 Report

I THANK the authors  for their work  and improvment of the manuscript  particularly  the introduction which support better the interest of the study.

there are  however still some comments.

1) 10 variables  in a logisitc regression with les Than 100 patients give usally instable models.

It should be of interest to give in the same table  UNivariate and multivariate analysis including in the multivariate model all variables which are significantly associated to the Antibody response (AR) at 6 months and some choosen varibles that are considered important by authors ( some comorbidities but not all) despite they are not associated to AR.

2) Authors have to precise that patients and Healthy controls who do not have any response at 4 weeks are not included in the model  because in a small number of patients they can influence the results significantly. If they  include theses patients or HC  they need to exclude them and do again the analysis.

3) if the model is true  with corrections, authors may conclude more strongly that their hypothesis of an impact of chronic disorder of consciousness on immune  response is not true but the this Chronic status impaired long term  independantly form age

Author Response

1) Following the reviewer's suggestion a new data analysis has been performed with univariate and multivariate approach. Please see the new table 3, where the additional regression model confirms the significant impact of Group and Age on antibody response.

2) Done. See statistical analysis paragraph

3) Following reviewer's suggestion we modified conclusions in the abstract and discussion.